# Individual identification of sika deer (*Cervus nippon*) using short tandem repeat analysis for investigating illegal carcass disposal in Japan

**Aki Tanaka** [ID]*◉, **Reina Ueda**◉, **Chihiro Udagawa**‡, **Toshinori Omi**‡, **Yuko Kihara** [ID]‡, **Shin-ichi Hayama**‡

Department of Veterinary Medicine, Nippon Veterinary and Life Science University, Tokyo, Japan

◉ These authors contributed equally to this work.
‡ These authors also contributed equally to this work.
* atanaka@nvlu.ac.jp

## Abstract

The population of Japanese sika deer (*Cervus nippon)* are controlled by hunting to prevent damage to various crops in many areas in Japan. Hunters are subsidized by submitting the tail to the local government; animal carcasses must be properly disposed of after the hunt, and abandonment of hunted deer in the field is prohibited by law. However, there have been many carcasses of sika deer being abandoned without proper disposal. In such cases, individual identification by DNA analysis is required to match the abandoned deer and submitted tail and identify the suspect. When identifying individual wildlife by DNA analysis, it is crucial to select appropriate markers that consider both the procedure of the analysis and the animal species. To evaluate availability of Short Tandem Repeat (STR) analysis for the identification of sika deer, this study aimed to construct an STR database for sika deer in Japan and to evaluate the discrimination power of STR markers, using an identification kit for a closely related species of cattle and STR markers of the sika deer. The results showed polymorphism at six STR loci from the Bovine Genotypes Panel 3.1 Kit and two STR markers for sika deer, suggesting that these loci may be useful for sika deer identification. The coincidence rate for the three STR loci (CSSM019, TGLA53, ETH10) was $7.63 \times 10^{-4}$, which was considered sufficient for identification of the sika deer population. This study was the first to evaluate the availability of sika deer STR analysis for individual identification in Japan and was expected to have applications in crime scene and wildlife forensics.

## Introduction

Wildlife forensics involves using various scientific disciplines to uphold laws and address illegal activities related to wildlife. This field includes taxonomy, pathology, molecular biology, biochemistry, genetics, and toxicology [1]. Apart from gathering evidence and prosecuting wildlife crimes, veterinary forensic science also aids in addressing a wide array of wildlife-related issues, such as monitoring changes in the environment and investigating disease

**Data availability statement:** Data is available from Dryad data repository. DOI: 10.5061/dryad.2547d7x2z.

**Funding:** The author(s) received no specific funding for this work.

**Competing interests:** The authors have declared that no competing interests exist.

epidemiology [1]. In situations where there may not be a distinct "victim" providing information, wildlife forensics is crucial in connecting suspects, victims, and crime scenes by analyzing trace and deteriorated physical evidence recovered from the crime scene [2].

Morphological, isotopic, and DNA analyses are employed to determine evidence [3]. Among these, DNA evidence holds significant value in identifying wildlife origin, especially when the evidence is already processed, degraded, or lacks identifiable morphology [4]. Advancements in DNA analysis techniques for use in wildlife crime investigations, known as 'wildlife DNA identification,' have been acknowledged for some time but are now increasingly recognized as a critical discipline [5].

Sika deer (*cervus nippon*) are native to Japan and eastern Asia, but are an invasive species with a global distribution, including 39 countries worldwide [6]. Sika deer live in forests and have a small home range, with males having a larger home range than females [7]. The species range has expanded elsewhere in Japan, causing damage to farmland, forestry and natural vegetation through soil erosion, especially in dense areas [8]. A number of reasons have been discussed for the overpopulation of sika deer in Japan, including reduction in human population, decrease in snow cover period, rapid increase in abandoned farmland, declining hunting population and an ageing population [6,9].

Due to the increase in the sika deer (*Cervus nippon*) population in Japan, most prefectures are managing the populations and encouraging extermination by providing subsidies to those who hunted the sika deer. Hunters are mandated to submit the tail of the dead deer to the local government office and exterminated animal carcasses must be handled properly by hunters, and it is forbidden to leave behind in the field. However, there have been increasing cases in Japan where exterminated animal carcasses had been abandoned without proper disposal and has become a serious legal issue. In investigation of such cases, determining the genotypes of the tails submitted to the local government and of the abandoned individuals, and investigating whether there are identical individuals that match, will provide beneficial information for the crime investigation and lead to the deterrence of illegal disposal. Individual identification through DNA analysis is therefore required and considered to be crucial to solve these cases.

In our previous study, mtDNA analysis was beneficial to identify individual sika deer as exclusionary evidence, however, complete identification was shown to be difficult (data not shown). Therefore, in this study, we focused on Short Tandem Repeat (STR) analysis, which is known to be the most discriminating genetic marker.

STRs are highly polymorphic, and analysis of multiple STR loci can provide improved discrimination. In forensic identification of individuals, a large number of studies have been conducted on individual markers, including sequence differences, mutation rates, and genetic characteristics among populations. Commercial kits are then used in which the marker sets are pre-mixed at appropriate concentrations so that all loci can be amplified uniformly [10]. When STR loci were first used in 1994, four STR loci were used, and later kits have allowed up to 24 STR loci to be used.

It is difficult to achieve same results in the very diverse animal populations encountered in wildlife medicine. Therefore, we focused on the most discriminating set of markers from the different STR sequences of the target species [11]. However, only a limited number of species have been characterized at targeted STR loci, and these are primarily for commercial or conservation genetic purposes rather than for forensic veterinary purposes [12]. Furthermore, test kits with established marker sets have been developed for domestic species of commercial interest, such as cattle, horses, and dogs, and thus cannot be used for individual identification in wildlife forensics [13]. Useful STR markers and testing protocols are also highly dependent on the degree of genetic differentiation between populations. It has been reported that even

the number of alleles in sika deer varies among populations, islands (regions), and mtDNA haplotype groups [14]. Therefore, it is necessary to develop and validate the usefulness of a unique set of markers for each local population to be evaluated.

The selection, development and analysis of STR markers is time-consuming and costly. However, in many STR analysis techniques, broad similarities in DNA sequences can provide useful information even with STR markers designed for closely related species. If heterologous primers can be used to amplify STR loci from different species within the same genus, family or order, the time and cost of isolating species-specific markers can be saved [15,16]. Such cross-species use of STR markers is the easiest way to obtain STR markers of different polymorphisms, and studies have been conducted in the deer family to verify their usefulness using markers from the closely related Bovidae [14,16–19]. These studies have shown that cattle and deer share extensive chromosomal homology [14,20], and it is expected that commercially available cattle identification kits could be used for sika deer.

In wildlife forensics, there is a need to develop large genetic databases to provide representative allele frequencies for populations and to provide quantitative probabilities for discrimination power through statistical analysis [5]. The nature of STR databases is species-specific, and STR typing systems vary widely from species to species, and these systems need to be developed, validated, and standardized for a large number of species [13]. STR allele frequency databases have been constructed for many domestic animal populations, including dogs, cats, and cattle [21–24], but few such databases exist for wildlife [25]. The databases that do exist are often based on a single population of a particular species [26], which can lead to large errors in the estimation of allele frequencies and discrimination power. Thus, common STR databases are not currently available for wildlife DNA assessment because standardized investigation and analysis of STR data is not performed except for livestock. Therefore, a new STR database should be established for species and local populations to be assessed [19].

Few markers sets and allele frequency databases have been developed for forensic genotyping in deer species, and most have been developed for deer species in North America [27]. Examples of deer populations for which marker sets and allele frequency databases have been developed include elk populations in California [28], white-tailed deer populations in northern New England [25], red deer populations in Hungary [29], red deer populations in England [30], and Japanese and British sika deer populations [14]. In Japan, marker sets have been developed in population structure analysis studies [31,32], but their usefulness in forensic veterinary practice has not been tested in any regional population. Therefore, it is necessary to validate the ability to identify individuals in Japan by constructing an STR database of Japanese deer and evaluating discrimination power by statistical analysis.

In this study, at the request of the law enforcement, individual identification by DNA analysis was conducted in an area where illegal disposal of sika deer had occurred, in order to determine the genotypes of the tails submitted to the government and of abandoned deer, and to investigate whether there were identical individuals that matched. "Bovine Genotypes Panel 3.1 Kit," a kit for the identification of bovine, a close relative of sika deer, and STR markers of sika deer were used to construct an STR database. Identification ability of the kits and markers used were also evaluated and examined for availability in wildlife forensics.

## Materials and methods

### Samples

Tails of sika deer that were submitted by hunters to the local government in 2022 in Shizuoka prefecture, Japan, for subsidy. Muscle fragment obtained from the tail samples that possessed same three mtDNA haplotypes (haplotype l-1, haplotype k-2, and haplotype h-2) as illegally

abandoned deer carcasses were further processed for DNA analysis. DNA sample was extracted from 25 mg muscle fragments to a final yield of 100 μL using a DNA extraction kit (DNeasy Blood & Tissue Kit, Qiagen, Venlo, The Netherlands) according to protocol. NanoDrop Lite (Thermo Fisher Scientific Inc., Walthman, MA, USA) was used to measure DNA concentration and purity. The extraction products were stored at −4°C until polymerase chain reaction (PCR).

## Amplification of the STR Locus by the PCR method

**Amplification of the STR locus using markers for closely related species.** Amplification of the STR locus of sika deer was first performed using the Bovine Genotypes Panel 3.1 Kit (Thermo Fisher Scientific Inc., Walthman, MA, USA), a kit provided for the parentage and individual identification of closely related bovine species. This kit is capable of co-amplifying 18 bovine STR loci (Table 1) in a single multiplex PCR and is fluorescently labeled with four fluorescent dyes (FAM, VIC, NED, PET).

The PCR composition was as follows: Primer mix 9 μl, master mix (buffer, deoxynucleoside triphosphates, Phusion Hot Start DNA Polymerase) 9 μl, template DNA (0.5 to 1 ng/μl) 2 μl. The extracted DNA samples were previously diluted with sterile water to a concentration of 0.5 to 1 ng/μl.

PCR amplification was performed on a T100 Thermal Cycler (Bio-Rad, Hercules, CA, USA) according to the manufacturer's recommended protocol: initial heat denaturation at 98°C for 1 minute, followed by 30 cycles of heat denaturation at 98°C for 20 seconds, annealing at 60°C for 75 seconds, and extension reaction at 72°C for 30 seconds, followed by final extension at 72°C for 5 minutes.

**Amplification of the STR locus using STR markers of Japanese deer.** Since few STR loci were amplified with the above kit, CSSM019 and BM6506 (Table 2), which were confirmed to be polymorphic in sika deer [32], were added and PCR was performed. Forward primers

**Table 1. Eighteen STR locus primers included in the Bovine Genotypes Panel 3.1 Kit. Adapted from the Bovine Genotypes Panel 3.1 Kit protocol.**

| Genetic locus | Chromosome[a] | Repeat base number[b] | Allele size (bp) | fluorescent dye |
|---|---|---|---|---|
| TGLA227 | 18 | di | 63–115 | FAM |
| BM2113 | 2 | di | 116–146 | FAM |
| TGLA53 | 16 | di | 147–197 | FAM |
| ETH10 | 5 | di | 198–234 | FAM |
| SPS115 | 15 | di | 240–270 | FAM |
| SPS113 | 10 | di | 279–307 | FAM |
| RM067 | 4 | di | 83–101 | VIC |
| TGLA126 | 20 | di | 104–132 | VIC |
| TGLA122 | 21 | di | 133–193 | VIC |
| INRA23 | 3 | di | 194–236 | VIC |
| BM1818 | 23 | di | 248–276 | VIC |
| ETH3 | 19 | di | 89–131 | NED |
| ETH225 | 9 | di | 132–166 | NED |
| BM1824 | 1 | di | 170–218 | NED |
| CSRM60 | 10 | di | 79–115 | PET |
| MGTG4B | 4 | di | 129–153 | PET |
| CSSM66 | 14 | di | 171–209 | PET |
| ILSTS006 | 7 | di | 277–309 | PET |

[a]Chromosome numbers and allele sizes at which each locus was detected were those for bovine.

[b]The di in the repeat base number indicates that the locus consists of repeats of a two-base repetitive sequence.

**Table 2. STR markers for sika deer used in this study. Adapted from Shimamura et al [33].**

| Genetic locus | Repetition Number of bases | Allele size (bp) | Fluorescent dye | References |
|---|---|---|---|---|
| CSSM019 | di | 146–164 | HEX | Moore et al. (1992) [34]. |
| BM6506 | di | 202–218 | HEX | Bishop et al. (1994) [35]. |

were labeled with the fluorescent dye HEX for the above STR markers, and PCR set up was performed according to KOD FX Neo (TOYOBO Co., Ltd., Osaka, Japan). The PCR conditions were initial heat denaturation at 94°C for 2 minutes, followed by 30 cycles of heat denaturation at 98°C for 10 seconds, annealing at 57°C for 30 seconds, and extension reaction at 68°C for 10 seconds. The PCR mix was prepared according to PCR mix 25 μl [1 x PCR Buffer for KOD Fx Neo, 0.4 mM dNTPs, 0.3 μM each primer, < 100 ng/25 μl template DNA, 1 U/25 μl KOD Fx Neo, distilled water].

## Fragment analysis

The amplified PCR products were subjected to fragment analysis by Phasmac, Inc. The obtained fragment analysis data were genotyped by reading the waveform of each locus using Peak Scanner Software version 2.0 (Applied Biosystems, USA). A single extremely high peak around the expected allele size was determined to be homozygous, and two peaks more than 2 bp apart were determined to be heterozygous. In addition, if peaks of similar height (within 300 differences in fluorescence intensity (hereafter RFU)) to the stutter peak were observed in TGLA53, ETH10, and MGTG4B, the peak with the larger size was read, as this difference in height can be variable. The STR loci in the kit were read according to the recommended protocol, mainly from 300 to 6000 RFU, and the STR markers BM6506 and CSSM019 in the Nymphalidae were read from 100 to 33,000 RFU as valid peaks. Since ETH10 and SPS113 often appeared lower and higher than the reference values, respectively, 120–6000 RFU and 300–16000 RFU were defined as valid peaks.

## Statistical analysis

The genotyping data were subjected to linkage disequilibrium tests using GENEPOP version 4.0.5 [36]. The test settings were as follows, Power: 10000, Batches: 100, and iterations per batch: 10000 parameters.

The Hardy-Weinberg equilibrium test with chi-square goodness of fit test was then performed using CERVUS version 3.0.7 [37]. In addition, the observed heterozygosity value, expected heterozygosity value, power of discrimination, accumulated discrimination performance, and combined match probability were calculated.

Herein, n is the number of individuals or total number of alleles genotyped at each locus, and $pi(pj)$ is the $i(j)$th allele frequency in the population of allele n.

**Observed heterozygosity (H$o$).** Percentage of individuals that are heterozygous in the tested population. It is a measure of polymorphism at a test locus and is used to assess genetic variation at the individual level.

$$Ho = 1 - \frac{\left(\text{Observed homozygous logarithms}\right)}{(n)}$$

**Expected heterozygosity (H$e$).** Theoretical probability of heterozygosity. It is defined as the probability that two alleles randomly selected from the population are different, and is calculated from the allele frequencies by the following formula

$$He = 1 - \sum_{i=1}^{n} pi^2$$

**Power of Discrimination (PD).** A scale for evaluating the usefulness of test loci in identifying individuals.

$$PD = 1 - \sum_{i=1}^{n} pi^4 - \sum_{j>i}^{n} 4 \left( pi\, pj \right)^2$$

**Accumulated Discrimination Performance (APD).** The discrimination ability of all the tested seating positions taken as a whole.

$$APD = 1 - \prod_{i=1}^{n} \left( 1 - PDi \right)$$

(PD$i$ is the $i$-th discriminating power in the population of locus n)

**Combined match probability (MP).** The probability that two separate individuals have the same genotype in the aggregate for all tested loci.

$$MP = \prod_{i=1}^{n} \left( 1 - PDi \right)$$

The significance level for the chained disequilibrium and Hardy-Weinberg equilibrium tests was set to 0.05.

## Ethics statement

Ethical approval for this study was obtained from Institutional Animal Care and Use Committee (approval number: 2024K-29).

## Results

### STR Analysis

In this study, 84 tail samples from sika deer were used for the STR analysis. The median DNA concentration yield for the PCR analysis was 98.95 (range 7.5–2025.2) ng/μl (Supporting information). Polymorphisms were amplified in 80/84 samples at TGLA53, 50/84 samples at ETH10, 81/84 samples at SPS113, 68/84 samples at CSRM60, 30/84 samples at MGTG4B, and 14/84 samples at CSSM66. Of the other 12 STR loci, no amplification was observed at eight STR loci (TGLA227, SPS115, RM067, TGLA122, INRA23, BM1818, BM1824, ILSTS006) and 4 STR loci (BM2113, TGLA126, ETH3, ETH 225) were monomorphic although amplification was observed.

Fragmentation analysis in STR analysis using sika deer STR markers showed amplification and polymorphism in all samples in both BM6506 and CSSM019. Fragment analysis was used to calculate the allele size range, allele, number of alleles, heterozygous, homozygous and allele frequency from the genotypes of the eight STR loci detected in 84 samples of Nymphaea (Table 3). Genotyping results for 84 sika deer were described in the Supporting information.

Polymorphisms were found in alleles ranging from a minimum of three types of alleles in SPS113 (304, 308, 310) to a maximum of 14 types in TGLA53 (176, 178, 182, 188, 190, 194, 196, 199, 201, 205, 207, 212, 218, 220), with an average of 6.875 types per STR gene. 6.875 different alleles were detected per locus. The lowest allele frequency was 296 alleles in BM6506 and the highest allele frequency was 310 alleles in SPS113.

**Table 3. Allele size range, alleles, number of alleles, heterozygous, homozygous, and allele frequency of eight STR loci detected in 84 sika deer samples, 2022, Japan.**

| Genetic locus | Allele size range | Allele | Number of Allele | Heterozygous | Homozygous | Allele frequency |
|---|---|---|---|---|---|---|
| CSSM019 | 138–154 | 138 | 32 | 22 | 5 | 0.19 |
| | | 140 | 16 | 14 | 1 | 0.10 |
| | | 142 | 16 | 14 | 1 | 0.10 |
| | | 147 | 63 | 41 | 11 | 0.38 |
| | | 152 | 2 | 2 | 0 | 0.01 |
| | | 154 | 39 | 31 | 4 | 0.23 |
| BM6506 | 194–296 | 194 | 11 | 11 | 0 | 0.07 |
| | | 196 | 35 | 27 | 4 | 0.21 |
| | | 200 | 2 | 2 | 0 | 0.01 |
| | | 204 | 16 | 14 | 1 | 0.10 |
| | | 206 | 73 | 53 | 10 | 0.43 |
| | | 210 | 30 | 22 | 4 | 0.18 |
| | | 296 | 1 | 1 | 0 | 0.01 |
| TGLA53 | 176–220 | 176 | 1 | 1 | 0 | 0.01 |
| | | 178 | 3 | 3 | 0 | 0.02 |
| | | 182 | 2 | 2 | 0 | 0.01 |
| | | 188 | 2 | 2 | 0 | 0.01 |
| | | 190 | 2 | 2 | 0 | 0.01 |
| | | 194 | 8 | 6 | 1 | 0.05 |
| | | 196 | 5 | 5 | 0 | 0.03 |
| | | 199 | 20 | 16 | 2 | 0.13 |
| | | 201 | 9 | 9 | 0 | 0.06 |
| | | 205 | 39 | 33 | 3 | 0.24 |
| | | 207 | 46 | 26 | 10 | 0.29 |
| | | 212 | 5 | 5 | 0 | 0.03 |
| | | 218 | 16 | 14 | 1 | 0.10 |
| | | 220 | 2 | 2 | 0 | 0.01 |
| ETH10 | 221–241 | 221 | 44 | 20 | 12 | 0.44 |
| | | 222 | 2 | 0 | 1 | 0.02 |
| | | 233 | 2 | 2 | 0 | 0.02 |
| | | 235 | 15 | 9 | 3 | 0.15 |
| | | 237 | 30 | 18 | 6 | 0.30 |
| | | 241 | 7 | 7 | 0 | 0.07 |
| SPS113 | 304–310 | 304 | 24 | 16 | 4 | 0.15 |
| | | 308 | 41 | 29 | 6 | 0.25 |
| | | 310 | 97 | 33 | 32 | 0.60 |
| CSRM60 | 104–120 | 104 | 71 | 17 | 27 | 0.52 |
| | | 108 | 26 | 12 | 7 | 0.19 |
| | | 110 | 2 | 0 | 1 | 0.01 |
| | | 116 | 33 | 5 | 14 | 0.24 |
| | | 120 | 4 | 2 | 1 | 0.03 |
| MGTG4B | 143–171 | 143 | 10 | 2 | 4 | 0.17 |
| | | 152 | 3 | 1 | 1 | 0.05 |
| | | 154 | 23 | 1 | 11 | 0.38 |
| | | 161 | 17 | 1 | 8 | 0.28 |
| | | 165 | 3 | 1 | 1 | 0.05 |

*(Continued)*

**Table 3.** (Continued)

| Genetic locus | Allele size range | Allele | Number of Allele | Heterozygous | Homozygous | Allele frequency |
|---|---|---|---|---|---|---|
| | | 167 | 2 | 0 | 1 | 0.03 |
| | | 171 | 2 | 0 | 1 | 0.03 |
| CSRM66 | 196–210 | 196 | 1 | 1 | 0 | 0.04 |
| | | 198 | 1 | 1 | 0 | 0.04 |
| | | 202 | 9 | 9 | 0 | 0.32 |
| | | 204 | 8 | 8 | 0 | 0.29 |
| | | 206 | 3 | 3 | 0 | 0.11 |
| | | 208 | 4 | 4 | 0 | 0.14 |
| | | 210 | 2 | 2 | 0 | 0.07 |

Analysis of the correlation of alleles at different STR loci showed linkage disequilibrium (LD) (p< 0.05) with BM6506 and ETH10 in SPS113 (Table 4). However, BM6506 and ETH10 did not show linkage disequilibrium. Therefore, SPS113 was excluded from the analysis of accumulated discrimination performance and combined match probability.

To calculate indices of polymorphism, H*o*, H*e*, probability of significance in the Hardy-Weinberg equilibrium test (p-value), and PD were calculated from the number of heterozygotes, homozygotes, and allele frequency detected at each STR locus (Table 5).

The median nA at each STR locus was 6.5 (range 3–14). The median H*o*, calculated from the number of detected heterozygotes, was 0.648 (range 0.265–1.000), and the median H*e*, calculated from the allele frequency, was 0.739 (range 0.559–0.829), with differences among STR loci. The p-value of the Hardy-Weinberg equilibrium test calculated from H*o* and H*e* could not be tested in MGTG4B and CSRM66 due to the small number of samples. STR loci with p<0.05 were BM6506 (p=0.017) and CSRM60 (p<0.0001). The median PD calculated from allele frequencies was 0.888 (range 0.736–0.940), and in order of highest PD were TGLA53, CSRM66, CSSM019, MGTG4B, BM6506, ETH10, CSRM60, and SPS113.

Excluding SPS113, which showed linkage disequilibrium, two STR loci (MGTG4B, CSRM66) for which the Hardy-Weinberg equilibrium test was not possible, and two STR loci (BM6506, CSRM60) for which deviation from Hardy-Weinberg equilibrium was observed, three STR loci (CSSM019, TGLA53, ETH10), the APD calculated from the discriminatory power of the three STR loci was $1 - 7.63 \times 10^{-4}$ (= 0.99923734). The APD, or the probability that any two individuals have different genotypes, using the 3STR locus was 99.92%. The MP for the three STR locus was $7.63 \times 10^{-4}$, and the reciprocal of this value resulted in a genotype match with a probability of approximately 1 in 1311.

Of the 84 samples used in this study, 50 samples could be genotyped at all three STR loci mentioned above, and one pair (#72 and #88) had a matched genotype (Supporting information).

## Discussion

### Verifications of the marker by statistical analysis

In this study, to verify the STR analysis for identification of sika deer, genotypes were determined in 84 samples using the Bovine Genotypes Panel 3.1 Kit, a kit for identification of cattle, and two STR markers for sika deer. As a result, polymorphism was observed at six of the 18 STR loci in the Bovine Genotypes Panel 3.1 Kit and at both two STR markers for sika deer, for a total of eight STR loci genotyped. Using this data, a statistical analysis was performed to evaluate the individual power of each STR locus where polymorphisms were observed.

**Table 4. Results of linkage disequilibrium tests for eight STR loci detected in 84 sika deer, 2022, Japan.**

| Genetic locus1 | Genetic locus1 | p-value |
|---|---|---|
| CSSM019 | BM6506 | 0.277 |
| CSSM019 | TGLA53 | 0.650 |
| BM6506 | TGLA53 | 0.462 |
| CSSM019 | ETH10 | 0.507 |
| BM6506 | ETH10 | 0.072 |
| TGLA53 | ETH10 | 0.275 |
| CSSM019 | SPS113 | 0.898 |
| BM6506 | SPS113 | 0.013 |
| TGLA53 | SPS113 | 0.501 |
| ETH10 | SPS113 | 0.014 |
| CSSM019 | CSRM60 | 0.818 |
| BM6506 | CSRM60 | 0.326 |
| TGLA53 | CSRM60 | 0.206 |
| ETH10 | CSRM60 | 0.159 |
| SPS113 | CSRM60 | 0.281 |
| CSSM019 | MGTG4B | 0.893 |
| BM6506 | MGTG4B | 0.071 |
| TGLA53 | MGTG4B | 0.132 |
| ETH10 | MGTG4B | 0.857 |
| SPS113 | MGTG4B | 0.748 |
| CSRM60 | MGTG4B | 0.249 |
| CSSM019 | CSSM66 | 0.400 |
| BM6506 | CSSM66 | 0.404 |
| TGLA53 | CSSM66 | 0.893 |
| ETH10 | CSSM66 | 1.000 |
| SPS113 | CSSM66 | 0.472 |
| CSRM60 | CSSM66 | 0.599 |
| MGTG4B | CSSM66 | N/A |

**Table 5. H*o*,H*e*, p-values of Hardy-Weinberg equilibrium tests, and PD for the eight STR loci detected in 84 sika deer, 2022, Japan.**

| Genetic Locus | nA[a] | H*o* | H*e* | p-value | PD |
|---|---|---|---|---|---|
| CSSM019 | 6 | 0.738 | 0.755 | 0.876 | 0.900 |
| BM6506 | 7 | 0.774 | 0.727 | 0.017 | 0.885 |
| TGLA53 | 14 | 0.788 | 0.829 | 0.090 | 0.949 |
| ETH10 | 6 | 0.560 | 0.695 | 0.235 | 0.852 |
| SPS113 | 3 | 0.481 | 0.559 | 0.211 | 0.738 |
| CSRM60 | 5 | 0.265 | 0.636 | <0.0001 | 0.807 |
| MGTG4B | 7 | 0.100 | 0.750 | NA[b] | 0.891 |
| CSRM66 | 7 | 1.000 | 0.804 | NA | 0.917 |

[a]All allele number

[b]Not analyzed

The results showed that the H$o$ ranged from 0.265 to 1.000 (mean: 0.588) and the H$e$ ranged from 0.559 to 0.829 (mean: 0.719). Of the two STR loci that showed deviations from Hardy-Weinberg equilibrium in this study, CSRM60 had a low value of H$o$, while BM6506 had a high value. The deviation from Hardy-Weinberg equilibrium has been mainly attributed to the lack of heterozygosity, especially because H$o$ is reduced compared to the H$e$ [15]. CSRM60 had a significant difference compared to the other STR loci, especially in H$o$ and H$e$, and this effect may have caused the deviation from Hardy-Weinberg equilibrium. On the other hand, BM6506 has a relatively high heterozygosity observation value, which does not suggest such an effect, and it was considered that it might not show deviation from Hardy-Weinberg equilibrium as further studies accumulated allele frequency data.

Compared to H$o$ and H$e$ calculated in population structure studies of Japanese sika deer in Japan, studies of the entire Japanese population have reported results showing that H$e$ ranges from 0.19 for the Nagasaki population to 0.60 for the Hyogo population [14]. The mean H$o$ and H$e$ of the Hokkaido and Chiba populations were reported to be 0.21 and 0.23, respectively [38], while the H$o$ and H$e$ of the Shizuoka population were reported to range from 0.172 to 0.754 and 0.217 to 0.815, respectively [32]. Compared to these previous studies, the H$o$ and H$e$ detected in the present Japanese deer population were relatively high. Heterozygosity is affected by reduced genetic diversity due to inbreeding and bottlenecks [39,40], and these effects had been suggested in previous studies. However, the relatively high heterozygosity observed in this study suggests that such effects are less likely than in previous studies. Because the effects of bottlenecks vary among regional populations, the effects of reduced genetic diversity should be investigated in each assessment and in the regional population under study when performing STR analyses.

The results of the linkage disequilibrium test showed linkage disequilibrium with BM6506 and ETH10 in SPS113. STR loci in linkage disequilibrium should be excluded because linkage disequilibrium is a phenomenon in which non-random correlation is observed between alleles of multiple STR loci, i.e., the frequency of a particular combination (genotype) of them may be significantly higher [41]. The results of the Hardy-Weinberg equilibrium test were non-null for two STR loci (MGTG4B and CSRM66) and p < 0.05 for two STR loci (BM6506 and CSRM60). The Hardy-Weinberg law is established in populations where there are no factors such as selection, movement of individuals, or mutation, where voluntary mating is established, and where the population is sufficiently large [42]. Markers that deviate from Hardy-Weinberg equilibrium indicate difficulty in detection and inheritance of the STR locus [5], so it is preferable to eliminate them. Based on the above, excluding SPS113, which showed linkage disequilibrium, and two STR loci (BM6506, CSRM60) and two STR loci (MGTG4B, CSRM66), which showed deviation from Hardy-Weinberg equilibrium and were not testable, three STR loci (CSSM019, TGLA 53, ETH10) were used to calculate the APD and MP.

The APD calculated from the PD of the three STR loci (CSSM019, TGLA53, and ETH10) was $1-7.63 \times 10^{-4}$ (= 0.99923734). For reference, in a previous study analyzing the identification performance of wildlife by STR analysis, the APDs of 386 moose and 360 white-tailed deer in Belarus were reported to be $1-3.00 \times 10^{-9}$ (0.999999997) and $1-7.00 \times 10^{-5}$ (0.999993), respectively [27], and the APD of 142 wild boars in Italy was reported to be $1-4.00 \times 10^{-10}$ (0.99999999996) [43]. Compared to these previous studies, the identification performance of sika deer in this study is somewhat low: the MP for the three STR loci (CSSM019, TGLA53, and ETH10) was $7.63 \times 10^{-4}$. However, MP of 0.01–0.0001 was considered reasonable [44], and the results in this study were within this range.

The MP is affected by the number of markers, the allele frequency and the number of samples. In this study, there were only three markers that could be used to calculate the final MP, suggesting that the probability was relatively high. In addition, because sika deer live in groups

with the matrilineal herd as the basic unit [45], it is possible that some of the hunted deer were from the same herd. The MP, can be underestimated in populations that include consanguineous individuals [46], and since a bias toward matrilineally consanguineous individuals was suggested in our previous study, the possibility that the probability was underestimated due to this effect could not be ignored. However, the MP for the mtDNA markers validated in our previous study was 20.1%, providing a higher individual identification power than mtDNA analysis in this study. The overall identification power and MP obtained in this study may be sufficient for some local populations, and providing the statistical values together with the number of individuals in the local populations will be valuable information for crime investigation.

## The Bovine Genotypes Panel 3.1 Kit for the identification of the sika deer population

Among the STR loci included in the Bovine Genotypes Panel 3.1 Kit, polymorphisms were observed at SPS113 [15] and TGLA53 [17] as in the previous study, and new polymorphisms were identified at MGTG4B and ETH10 [16], which were amplified but monomorphic in the previous study. Polymorphisms were also observed at two other STR loci (CSSM66 and CSRM60), suggesting homology at these six STR loci.

However, the success rate of genotyping using the Bovine Genotypes Panel 3.1 Kit in this study was low: amplification was observed in less than half of all samples for MGTG4B and CSSM66, and for other STR loci amplification was confirmed in all samples. Since amplification was confirmed in all samples for the two sika deer STR markers, it is possible that the conditions of this kit were not optimal for application to sika deer. Therefore, the six STR loci confirmed to be polymorphic in this study could be used as a single STR marker instead of using the kit, which would increase the success rate of amplification. In addition, because the kit uses a two-base repeat STR locus, the ratio of stutter peaks was high, making genotype discrimination difficult, especially for TGLA53, ETH10, and MGTG4B. The two-base repeat STR locus is commonly used in animals, especially in livestock [13], which complicates data interpretation by making sample contamination and heterozygosity interpretation difficult. Individual identification by STR analysis in humans uses repeat sequences of four to five bases, which have very few stutter peaks and clear separation between alleles [5].

Although previous studies have suggested a decrease in genetic diversity of the two-tailed deer due to bottlenecks, the present study showed that the effect was low, and that haplotype and allele frequencies varied among local populations. Therefore, it was considered necessary to select markers appropriate for the local populations to be tested and to determine the degree of genetic diversity when conducting DNA identification in wildlife forensic veterinary medicine. Although further accumulation of data and expansion of markers with higher discriminatory power are needed in the future, this is the first study to evaluate the usefulness of STR analysis for individual identification for sika deer in Japan and it is expected to be applied to individual identification in wildlife forensic veterinary medicine. Useful marker sets should be developed, and databases should be constructed for other animal species in Japan as well for the future investigation.

## Limitations of the study

The low success rate of genotyping in this study could be due to low DNA concentration and/or DNA degradation; low DNA concentration increases the risk of allele dropout and overall amplification failure [47]. DNA yield may be low and degraded, particularly in wildlife forensics, since the samples are commonly decayed and not necessarily optimal for DNA

analysis. Therefore, if fresh samples were used, the success rate of genotyping would be higher and achieve better statistical values. Wildlife forensics samples are often degraded and thus a higher risk of such genotyping failures should be considered when performing STR analysis.

## Conclusion

In this study, STR analysis was used to genotype the sika deer and to evaluate its usefulness for individual identification. A total of eight STR loci were genotyped from the bovine individual identification kit and the bison STR marker, and statistical analysis was performed using these allele frequency data. The results showed that the number of alleles and heterozygosity were generally high, although there were differences among the STR loci, suggesting that the sika deer population used in this study was less affected by reduced genetic diversity due to bottlenecks. Finally, the overall PD and MP were evaluated using the three STR loci, excluding the STR loci that were in linkage disequilibrium and out of Hardy-Weinberg equilibrium, which were $1–7.63 \times 10^{-4}$ (= 0.99923734) and $7.63 \times 10^{-4}$, respectively. These results indicated that the marker sets used in this study had sufficient discrimination power, depending on the number of individuals in each local population, and can be applied to wildlife DNA identification and would provide beneficial evidence for crime investigation. Since several homologous markers were obtained in the bovine identification kit, it is expected that further research will improve the ability to identify individuals by using more single STR markers, which will be beneficial in wildlife forensic.

## Supporting information

**S1 Fig. Example of analysis of STR loci and CSSM019 and BM6506 of amplified bovine genotypes Panel 3.1 Kit.**
(TIFF)

**S1 File. Genotyping results of 84 sika deer, 2022, Japan.**
(XLSX)

**S2 File. Concentration of DNA (ng/ul) for PCR analysis in 84 sika deer, 2022, Japan.**
(XLSX)

## Acknowledgments

We wish to thank the timely help given by the Laboratory of Wildlife Medicine at Nippon Veterinary and Life Science University in analyzing the large number of samples.

## Author contributions

**Conceptualization:** Aki Tanaka, Reina Ueda, Toshinori Omi, Yuko Kihara, Shin-ichi Hayama.

**Data curation:** Aki Tanaka, Reina Ueda, Chihiro Udagawa, Toshinori Omi, Yuko Kihara, Shin-ichi Hayama.

**Formal analysis:** Aki Tanaka, Reina Ueda, Chihiro Udagawa, Toshinori Omi, Yuko Kihara.

**Investigation:** Aki Tanaka, Reina Ueda, Chihiro Udagawa, Toshinori Omi, Yuko Kihara, Shin-ichi Hayama.

**Methodology:** Aki Tanaka, Reina Ueda, Chihiro Udagawa, Toshinori Omi, Yuko Kihara, Shin-ichi Hayama.

**Project administration:** Aki Tanaka.

**Resources:** Aki Tanaka.

**Software:** Aki Tanaka.

**Supervision:** Aki Tanaka, Toshinori Omi, Shin-ichi Hayama.

**Validation:** Aki Tanaka, Shin-ichi Hayama.

**Visualization:** Aki Tanaka.

**Writing – original draft:** Aki Tanaka, Reina Ueda, Yuko Kihara, Shin-ichi Hayama.

**Writing – review & editing:** Aki Tanaka, Reina Ueda, Chihiro Udagawa, Toshinori Omi, Yuko Kihara, Shin-ichi Hayama.

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
