## [Decision Letter · Decision Letter 0]

16 Oct 2024

PONE-D-24-33085Individual identification of sika deer (Cervus nippon) by Short Tandem Repeat (STR) analysis for illegal disposal case in JapanPLOS ONE

Dear Dr. Tanaka,

Thank you for submitting your manuscript to PLOS ONE. After careful consideration, we feel that it has merit but does not fully meet PLOS ONE’s publication criteria as it currently stands. Therefore, we invite you to submit a revised version of the manuscript that addresses the points raised during the review process.

Please attend to all of the required changes outlined in the Editor's comment below before resubmission

We look forward to receiving your revised manuscript.

Kind regards,

Jo-Ann L. Stanton, Ph.D

Academic Editor

PLOS ONE

3. We note you have included a table to which you do not refer in the text of your manuscript. Please ensure that you refer to Table 1 in your text; if accepted, production will need this reference to link the reader to the Table.

Additional Editor Comments:

Please correct the table referencing as it is incorrect e.g. Table 1 is in the title but Table 7 is referred to in the text.

Use dye name in table 1, not the colour.

Ethics statement – what institution approved this work?

From Line 321 - State ranges in brackets = (range x – xx) not as as range (x – xx).

From Line 334 – Please correct the bracketting errors

Line 441 – An arguement is make for low DNA concentration affecting the quality of the STR calls but concentration data is not referred to in the results section. The data needs to be highlighted if this is to form part of the discussion. Please provide or mention this in the results section

Please make sure the Supp. Data is referred to in the main manuscript text.

Improve figure resolution.

Also address the following comments from Reviewer 1:

L29 : cases can change to carcasses (Cases it may confuse) update

L 154: The protocol should be properly cited. The mention of "Thermo Fisher Scientific Inc." is vague and should be clarified with appropriate reference.

L175-179: Consider including these details as a note beneath Table 1.

Statistical Analysis: Ensure that all mathematical notations are formatted in italic style throughout the manuscript.

Results:

The tables should be formatted consistently according to the journal’s guidelines.

Numerical data should be presented in a scientific format or limiting values to three decimal places both in the text and in the tables to maintain consistency.

The figures are of insufficient quality; the data is unreadable. Please provide high-resolution images.

Reviewers' comments:

Reviewer's Responses to Questions

**Comments to the Author**

1. Is the manuscript technically sound, and do the data support the conclusions?

Reviewer #1: Yes

Reviewer #2: Yes

2. Has the statistical analysis been performed appropriately and rigorously? 

Reviewer #1: Yes

Reviewer #2: Yes

3. Have the authors made all data underlying the findings in their manuscript fully available?

Reviewer #1: Yes

Reviewer #2: Yes

4. Is the manuscript presented in an intelligible fashion and written in standard English?

Reviewer #1: Yes

Reviewer #2: Yes

5. Review Comments to the Author

Reviewer #1: In this study, the authors investigated the individual identification of sika deer (Cervus nippon) using Short Tandem Repeat (STR) analysis to address the illegal disposal of carcasses in Japan. The research presents a compelling approach that could be applied to the illegal disposal of carcasses of other species as well. However, several significant revisions are necessary before the manuscript is suitable for publication. Below are some representative comments:

Title:

Acronyms should be avoided in the title. A more refined version could be: "Individual Identification of Sika Deer (Cervus nippon) Using Short Tandem Repeat Analysis for Monitoring Illegal Carcass Disposal in Japan." (This is just a reference you can upgrade better than this)

Authors affiliation should formatted as Journal style.

Abstract is too long, update in single paragraph

L29 : cases can change to carcasses (Cases it may confuse) update

Introduction

Introduction:

The introduction is well-written and effectively outlines the study's relevance. To enhance the broader context, it is recommended to add a paragraph discussing the distribution and common behavior of sika deer both locally and regionally, referencing recent studies such as:

Galon, E. M. S. (2023). Molecular epidemiological studies on livestock tick-borne parasitic diseases in the Philippines.

Dhakal, T., Kim, T. S., Kim, S. H., Tiwari, S., Kim, J. Y., Jang, G. S., & Lee, D. H. (2023). Distribution of sika deer (Cervus nippon) and the bioclimatic impact on their habitats in South Korea. Scientific Reports, 13(1), 19040.

Kaji, K., Uno, H., & Iijima, H. (2022). Future Challenges for Research and Management of Sika Deer. In Sika Deer: Life History Plasticity and Management (pp. 615-634). Singapore: Springer Nature Singapore.

Dhakal, T., Jang, G. S., Kim, M., Kim, J. H., Park, J., Lim, S. J., ... & Lee, D. H. (2023). Habitat utilization distribution of sika deer (Cervus nippon). Heliyon, 9(10).

Material and methods

L 154: The protocol should be properly cited. The mention of "Thermo Fisher Scientific Inc." is vague and should be clarified with appropriate reference.

L175-179: Consider including these details as a note beneath Table 1.

Statistical Analysis: Ensure that all mathematical notations are formatted in italic style throughout the manuscript.

Results:

The tables should be formatted consistently according to the journal’s guidelines.

Numerical data should be presented in a scientific format or limiting values to three decimal places both in the text and in the tables to maintain consistency.

The figures are of insufficient quality; the data is unreadable. Please provide high-resolution images.

Discussion and conclusion

While the authors have contextualized their findings within the framework of previous research, the manuscript lacks a discussion of future research directions. It is recommended to include potential avenues for further investigation.

Good luck

Reviewer #2: The paper provide important information which is useful in identification of sika deer (Cervus nippon) using molecular markers as STRs. The paper is well written and all the technical aspects were covered.

6. PLOS authors have the option to publish the peer review history of their article (what does this mean? ). If published, this will include your full peer review and any attached files.

**Do you want your identity to be public for this peer review?** For information about this choice, including consent withdrawal, please see our Privacy Policy .

Reviewer #1: No

Reviewer #2: No

---

## [Author Response · Author response to Decision Letter 1]

5 Feb 2025

Thank you for your reviewing our manuscript “Individual identification of sika deer (Cervus nippon) by Short Tandem Repeat (STR) analysis for illegal disposal case in Japan”. I apologize for the length of time it has taken to revise the manuscript due to my health issues. We appreciate your time and effort to improve our manuscript. We would like to address each of the reviewers' comments. We have submitted our manuscript to professional English editing agency before submission.

Answer: I had revised according to PLOS ONE’s style requirements.

Answer: I have changed the data availability statement so that we can share entire data freely before acceptance.

3. We note you have included a table to which you do not refer in the text of your manuscript. Please ensure that you refer to Table 1 in your text; if accepted, production will need this reference to link the reader to the Table.

Answer: I apologize for the confusion. I have changed Table 7 to Table 1 in Line 164.

Answer: I apologize again for the confusion. I have revised the reference number in Table 2, [120] to [33], [121] to [34], and [122] to [35], and reformatted the Reference section.

Additional Editor Comments:

Please correct the table referencing as it is incorrect e.g. Table 1 is in the title but Table 7 is referred to in the text.

Answer: I apologize for the confusion. I have changed Table 7 to Table 1 in Line 164.

Use dye name in table 1, not the colour.

Answer: I have changed blue to FAM, green to VIC, black to NED and red to PET in Table 1.

Ethics statement – what institution approved this work?

Answer: This work was approved by Nippon Veterinary and Life Science University.

From Line 321 - State ranges in brackets = (range x – xx) not as as range (x – xx).

Answer: I have changed the brackets to (range x-xx).

From Line 334 – Please correct the bracketing errors

Answer: I have corrected the bracketing error to (CSSM019, TGLA53, ETH10) in Line 338.

Line 441 – An argument is make for low DNA concentration affecting the quality of the STR calls but concentration data is not referred to in the results section. The data needs to be highlighted if this is to form part of the discussion. Please provide or mention this in the results section

Answer: I added the median DNA concentration and the range for our PCR analysis in the Result section as “The median DNA concentration yield for the PCR analysis was 98.95 (range 7.5-2025.2) ng/μl (Supporting information)”, Line 274-275 and also provided individual measurements in Supporting information. I have also added “S2 File. This is the S1 File Concentration of DNA (ng/ul) for PCR analysis in 84 sika deer, 2022, Japan.” in Line 652-653.

Please make sure the Supp. Data is referred to in the main manuscript text.

Answer: I added “Genotyping results for 84 sika deer were described in the Supporting information.” in the Result section, Line 285-286.

Improve figure resolution.

Answer: I apologize for low resolution. In order to improve figure resolution, I reduced the number of graphs and presented cases CSSM019 and BM6506 which showed high peaks. I changed the figure title to “This is the S1 Fig Example of analysis of STR loci and CSSM019 and BM6506 of amplified bovine genotypes Panel 3.1 Kit” in Line 627-628.

Comments from Reviewer 1:

Title:

Acronyms should be avoided in the title. A more refined version could be: "Individual Identification of Sika Deer (Cervus nippon) Using Short Tandem Repeat Analysis for Monitoring Illegal Carcass Disposal in Japan." (This is just a reference you can upgrade better than this)

Answer: Thank you for your suggestion. I changed the title to “Individual identification of sika deer (Cervus nippon) using short tandem repeat analysis for investigating illegal carcass disposal in Japan” in Line 4 and 5.

Authors affiliation should format as Journal style.

Answer: I revised the authors’ affiliation as Journal style.

Abstract is too long, update in single paragraph

Answer: I shortened the abstract and updated in a single paragraph.

L29 : cases can change to carcasses (Cases it may confuse) update

Answer: I revised to “carcasses” in Line 26.

Introduction:

The introduction is well-written and effectively outlines the study's relevance. To enhance the broader context, it is recommended to add a paragraph discussing the distribution and common behavior of sika deer both locally and regionally, referencing recent studies such as:

Galon, E. M. S. (2023). Molecular epidemiological studies on livestock tick-borne parasitic diseases in the Philippines.

Dhakal, T., Kim, T. S., Kim, S. H., Tiwari, S., Kim, J. Y., Jang, G. S., & Lee, D. H. (2023). Distribution of sika deer (Cervus nippon) and the bioclimatic impact on their habitats in South Korea. Scientific Reports, 13(1), 19040.

Kaji, K., Uno, H., & Iijima, H. (2022). Future Challenges for Research and Management of Sika Deer. In Sika Deer: Life History Plasticity and Management (pp. 615-634). Singapore: Springer Nature Singapore.

Dhakal, T., Jang, G. S., Kim, M., Kim, J. H., Park, J., Lim, S. J., ... & Lee, D. H. (2023). Habitat utilization distribution of sika deer (Cervus nippon). Heliyon, 9(10).

Answer: Thank you for providing references. I added a paragraph describing sika deer population in Introduction section “Sika deer (cervus nippon) are native to Japan and eastern Asia, but are an invasive species with a global distribution, including 39 countries worldwide[6]. Sika deer live in forests and have a small home range, with males having a larger home range than females[7]. The species range has expanded elsewhere in Japan, causing damage to farmland, forestry and natural vegetation through soil erosion, especially in dense areas[8]. A number of reasons have been discussed for the overpopulation of sika deer in Japan, including reduction in human population, decrease in snow cover period, rapid increase in abandoned farmland, declining hunting population and an ageing population[6, 9].” in Line 58-66.

Material and methods

L 154: The protocol should be properly cited. The mention of "Thermo Fisher Scientific Inc." is vague and should be clarified with appropriate reference.

Answer: I apologize for our incomplete explanation. I added “NanoDrop Lite (Thermo Fisher Scientific Inc., Walthman, MA, USA) was used to measure DNA concentration and purity.” in the Method section in Line 155-156.

L175-179: Consider including these details as a note beneath Table 1.

Answer: I moved “Adapted from the Bovine Genotypes Panel 3.1 Kit protocol.

Chromosome numbers and allele sizes at which each locus was detected were those for bovine.

The di in the repeat base number indicates that the locus consists of repeats of a two-base repetitive sequence.” as a note beneath Table 1 in Line 178-182.

Statistical Analysis: Ensure that all mathematical notations are formatted in italic style throughout the manuscript.

Answer: I formatted all the mathematical notations in italic style throughout the manuscript.

Results:

The tables should be formatted consistently according to the journal’s guidelines.

Answer: I formatted all the tables according to the journal’s guidelines.

Numerical data should be presented in a scientific format or limiting values to three decimal places both in the text and in the tables to maintain consistency.

Answer: I apologize for the confusion. Except for values that interfere with the interpretation of the result, numerical data are revised to three decimal places both in the text and in the tables.

The figures are of insufficient quality; the data is unreadable. Please provide high-resolution images.

Answer: I apologize for the low resolution. In order to improve figure resolution, I reduced the number of graphs and presented cases CSSM019 and BM6506 which showed high peaks. I changed the figure title to “This is the S1 Fig Example of analysis of STR loci and CSSM019 and BM6506 of amplified bovine genotypes Panel 3.1 Kit” in Line 627-628.

Discussion and conclusion

While the authors have contextualized their findings within the framework of previous research, the manuscript lacks a discussion of future research directions. It is recommended to include potential avenues for further investigation.

Answer: Thank you for your suggestion. I have added “Although previous studies have suggested a decrease in genetic diversity of the two-tailed deer due to bottlenecks, the present study showed that the effect was low, and that haplotype and allele frequencies varied among local populations. Therefore, it was considered necessary to select markers appropriate for the local populations to be tested and to determine the degree of genetic diversity when conducting DNA identification in wildlife forensic veterinary medicine. Although further accumulation of data and expansion of markers with higher discriminatory power are needed in the future, this is the first study to evaluate the usefulness of STR analysis for individual identification for sika deer in Japan. and it is expected to be applied to individual identification in wildlife forensic veterinary medicine. Useful marker sets should be developed, and databases should be constructed for other animal species in Japan as well for the future investigation.” in the Discussion section, Line 444-454.

---

## [Editor Report · Decision Letter 1]

28 Feb 2025

PONE-D-24-33085R1Individual identification of sika deer (Cervus nippon) using short tandem repeat analysis for investigating illegal carcass disposal in JapanPLOS ONE

Dear Dr. Tanaka,

Thank you for submitting your manuscript to PLOS ONE. After careful consideration, we feel that it has merit but does not fully meet PLOS ONE’s publication criteria as it currently stands. Therefore, we invite you to submit a revised version of the manuscript that addresses the points raised during the review process. The manuscript has addressed both mine and the previous reviewers comments but presentation and typographical errors remain.  Some are presented below but please carefully review any resubmission for these types of error to avoid further delay.

We look forward to receiving your revised manuscript.

Kind regards,

Jo-Ann L. Stanton, Ph.D

Academic Editor

PLOS ONE

Journal Requirements:

Additional Editor Comments:

Please revise the text for typographical errors. These must be corrected to meet the presentation standards of PLOS One.

Table 1: please review the journal standards for annotations to tables

Line 150: Subsidy not Subsid.

Line 214: Remove the sentence "The PCR was carried out as follows'. No instructions follow.

Line 390: Italics error for He

Line 396: Italics error for He and Ho

Abbreviations: If you define an abbreviation in the text e.g. PD, use the abbreviation. Define the term the first time it appears in the paper and use only the abbreviation from then on. If a term is only used once, do not give it an abbreviation.

Round all numbers to three decimal places: see line 421, 424 and 425.

Line 473: Remove the full stop from this sentence "for sika deer in Japan. and it is expected to be ". It should read "...for sika deer in Japan and it is expected to be...."

---

## [Author Response · Author response to Decision Letter 2]

2 Mar 2025

Thank you for reviewing our manuscript “Individual identification of sika deer (Cervus nippon) by Short Tandem Repeat (STR) analysis for illegal disposal case in Japan”. We appreciate your time and effort to improve our manuscript, and we apologize that we failed to correct the presentation and typographical errors. We carefully reviewed our manuscript and addressed all the comments from the Academic Editor.

Journal Requirements:

Answer: We have carefully reviewed our references and corrected the reference number in Line 426 and 441.

Additional Editor Comments:

Please revise the text for typographical errors. These must be corrected to meet the presentation standards of PLOS One.

Answer: I apologize for the typographical errors. We corrected them to meet the presentation standards of PLOS One.

Table 1: please review the journal standards for annotations to tables

Answer: We added footnotes in the Table 1 as follows “aChromosome numbers and allele sizes at which each locus was detected were those for bovine.

bThe di in the repeat base number indicates that the locus consists of repeats of a two-base repetitive sequence.” in Line 179-182.

”

Line 150: Subsidy not Subsid.

Answer: I apologize again for our typographical error. I corrected to “subsidy” in Line 150.

Line 214: Remove the sentence "The PCR was carried out as follows'. No instructions follow.

Answer: We removed the sentence “The PCR was carried out as follows” in Line 194. I apologize for the confusion.

Line 390: Italics error for He

Answer: We changed to Italic as He.

Line 396: Italics error for He and Ho

Answer: We changed to Italic as He and Ho.

Abbreviations: If you define an abbreviation in the text e.g. PD, use the abbreviation. Define the term the first time it appears in the paper and use only the abbreviation from then on. If a term is only used once, do not give it an abbreviation.

Answer: We have revised all the abbreviations throughout the manuscript, including PD, APD. MP, Ho and He.

Round all numbers to three decimal places: see line 421, 424 and 425.

Answer: I apologize that we did not round up to three decimal places. The difference between those numbers were in the 9th, 5th and 10th decimal points and if we round them up in three decimal places, we would not be able to show the difference of the identification performances in the previous studies. Therefore, we included calculation formula as follow; “1-3.00 × 10-9 (0.999999997), 1-7.00 × 10-5 (0.999993) and 1-4.00 × 10-10 (0.99999999996)” in Line 400, 403 and 404.

Line 473: Remove the full stop from this sentence "for sika deer in Japan. and it is expected to be ". It should read "...for sika deer in Japan and it is expected to be...."

Answer: We apologize for the typographical error. I changed to “for sika deer in Japan and it is expected to be” in Line 452.

---

## [Editor Report · Decision Letter 2]

5 Mar 2025

Individual identification of sika deer (Cervus nippon) using short tandem repeat analysis for investigating illegal carcass disposal in Japan

PONE-D-24-33085R2

Dear Dr. Tanaka,

We’re pleased to inform you that your manuscript has been judged scientifically suitable for publication and will be formally accepted for publication once it meets all outstanding technical requirements.

Kind regards,

Jo-Ann L. Stanton, Ph.D

Academic Editor

PLOS ONE
---

## [Editor Report · Acceptance letter]

PONE-D-24-33085R2

PLOS ONE

Dear Dr. Tanaka,

I'm pleased to inform you that your manuscript has been deemed suitable for publication in PLOS ONE. Congratulations! Your manuscript is now being handed over to our production team.

Kind regards,

on behalf of

Dr. Jo-Ann L. Stanton

Academic Editor

PLOS ONE